# Physical Metallurgical Bonding Layer Formed between Fe$_{80}$Si$_9$B$_{11}$ Metallic Glass and Crystalline Aluminum in Rolled Composite Plate by High-Pressure Torsion at Room Temperature

**Shengfeng Shan [1],\*, Xiaopeng Zhang [2], Haibo Guo [1] and Yuanzhi Jia [2],\***

[1] School of Industry, Jining University, Qufu 273155, China
[2] State Key Laboratory of Metastable Materials Science and Technology, Yanshan University, Qinhuangdao 066004, China
\* Correspondence: sdjnssf@jnxy.edu.cn (S.S.); ysujyz@126.com (Y.J.)

**Abstract:** Metallic glasses (MGs) have excellent properties, such as high strength and low elastic modulus, can be used as reinforcement in metal matrix composites. In this paper, aluminum matrix composites reinforced with Fe$_{80}$Si$_9$B$_{11}$ MG strips with different weight contents (5, 10, 15, 20 and 25%) were produced by roll-bonding at an initial temperature of 450 °C and 80% deformation. Tensile mechanical tests showed that the tensile strength of the composite sheets containing 10% MG strips showed the highest tensile strength of 166 MPa. Further studies on the sandwich structured samples were conducted using high-pressure torsion (HPT) technology with various pressures of 0.55 GPa, 1.10 GPa, 1.65 GPa, and 2.20 GPa. X-ray diffractometry (XRD), scanning electron microscopy (SEM), TriboIndenter nanomechanical testing, and transmission electron microscopy (TEM) were used to study the microstructures, mechanical properties and the bonding interface of the material. The results show that the hardness near the interface presented a transition area. High-resolution TEM observation showed that physical metallurgical bonding can be achieved between MG and aluminum alloy. A preliminary fitting of metallurgical bonding conditions was carried out according to the experimental parameters of HPT and the interface bonding condition in this study.

**Keywords:** roll-bonding; MG; high-pressure torsion; physical metallurgical bonding





## 1. Introduction

Metallic glasses (MGs) are a family of materials with superior properties such as high strength, high hardness [1–3], and good soft and hard magnetic properties [4–6], thus are promising for engineering applications [7–9]. However, owing to the critical fabrication conditions of MGs, their product sizes are rather limited. By joining them to common crystalline alloys, MGs can find more applications [10]. Moreover, when they are used as reinforcements to alloys, owing to their metallic nature, MGs could have better interfaces with the metallic matrix compared to those of inorganic materials such as ceramics and fibers [11]. Therefore, the bonding of MGs to crystalline metals has attracted increasing attentions [12]. Over the past few decades, various methods have been used to join MGs to MGs and MGs to crystalline alloys, including electron beam welding [13], friction welding [14], diffusion welding [15], and explosive welding of MG to aluminum [16]. Some of the methods were conducted at elevated temperatures of the supercooled liquid region of MGs, while some require particular technologies, which makes the joining of MGs to crystalline metals inconvenient. Therefore, a relatively simple method to bond MGs to crystalline alloys at room temperature is of interest to facilitate the application of MGs. Chen et al. investigated the effects of the main working parameters, including the temperature, pressure, and deformation strain, and reported that plastic deformation (PD)

has an important role in the process of diffusion bonding [15]. Regarding this, bonding MGs to crystalline alloys using severe PDs at room temperature can be considered. In this study, Al matrix composites with MG strips as the reinforcement were first fabricated by employing the roll bonding method, and the microstructures and mechanical properties of the composites were studied. The high-pressure torsion (HPT) method was further used to study the bonding between the MG strip and crystalline aluminum.

## 2. Materials and Methods

A 1060 industrial pure aluminum sheet and Fe-based amorphous alloy foil strips were used as raw materials for the roll bonding and HPT studies. The MG strips used in this study were $Fe_{80}Si_9B_{11}$, with a thickness of approximately 35 μm, purchased from Advanced Technology & Materials Co., Ltd., Beijing, China. The 2 mm-thick commercial 1060 sheet contains 0.25 wt. % Si, 0.03 wt. % Fe, Mg, and Ti, and 0.05 wt. % Zn and Cu. The dimensions of the roll bonding samples were 30 mm × 100 mm. Before rolling, the surfaces of the aluminum plate and Fe-based amorphous alloy foil strip were treated separately to ensure that the contact surface was free of pollution. The iron-based amorphous alloy strip was sandwiched between two layers of Al sheet, and hot-rolled at 450 °C at a rolling speed of 0.6 m/s, and the composite were obtained with a total deformation of 80%. The HPT samples (newly modified) were cut into a disc with a diameter of 12 mm using a spark cutting machine. The HPT experiment was carried out according to Ref. [10]. The sample was placed into the disc-shaped cavity of the anvil mold. The torsion strain was obtained by the rotation of the lower anvil mold. The surfaces of each disc were ultrasonically degreased before the HPT process. The working pressure was 0.5 to 2.5 GPa, while the maximum torsion strain in the sample was 5.5, at an angular speed of $0.017 \text{ s}^{-1}$. The microstructures of the as-processed samples were observed by transmission electron microscopy (TEM) and high-resolution transmission electron microscopy (HRTEM). The nanohardnesses of the samples were measured using a triboindenter employing a Berkovic tip, with a loading of 0.5 mN/s and a maximum load of 5 mN.

The deforming strain of the sample was calculated by [17]:

$$\varepsilon = \ln\left(2\pi \cdot n \cdot r \cdot h_0 / h^2\right) \tag{1}$$

where $n$ is the number of torsion rounds, $r$ is the radius, and $h_0$ and $h$ are the thicknesses of the sample before and after deformation, respectively.

## 3. Results

*Roll Bonding*

XRD curves of the MG strips examined intentionally from the composite material synthesized at the rolling temperature of 450 °C are shown in Figure 1. It can be seen that no obvious sharp diffraction peak is observed in the diffraction spectrum of this sample, which indicates that no crystalline or quasi-crystalline substances are formed in the MG after heating, thermal insulation, and mechanical rolling. The $Fe_{80}Si_9B_{11}$ MG strips remained amorphous during the bonding process, as did the MG strips before the roll bonding process.

Figure 2 shows the tensile strength test results of the composite plate material after roll bonding at an initial temperature of 450 °C and total deformation of 80% with different contents of MG (5%, 10%, 15%, 20%, 25%). The tensile strength of the rolled industrial pure aluminum was lowest, while its ultimate tensile strength was 102 MPa. The tensile strength of the rolled composite plate containing MG was improved to different degrees according to the content of MG in the aluminum alloy matrix. When the content of MG was increased from 0% to 10% through 5%, the tensile strength of the sample was gradually enhanced. The ultimate tensile strengths of the rolled composite plates with contents of MG of 5% and 10% were 131 and 166 MPa, respectively. When the content of MG was increased from 15% to 25% through 20%, the tensile strength of the sample was gradually weakened, while

the ultimate tensile strength was 153, 143, and 131 MPa, respectively. When the content of MG was approximately 10%, the ultimate tensile strength of the rolled composite plate was largest.

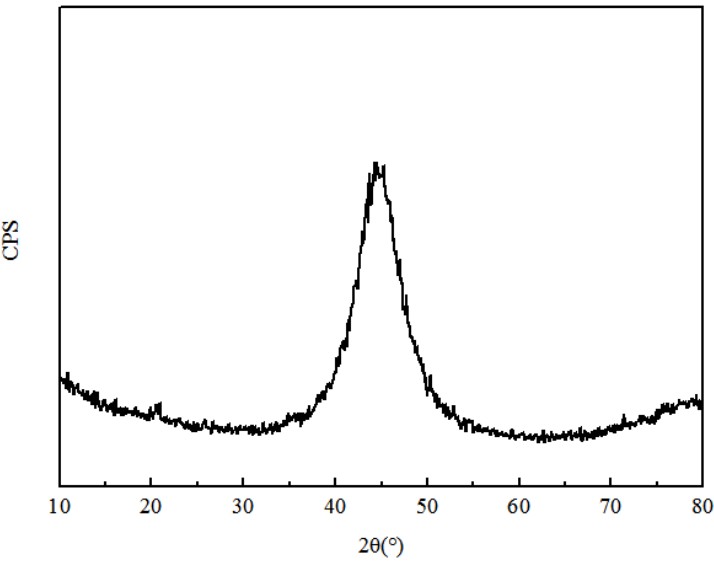

**Figure 1.** XRD curves of Fe80Si9B11 MG in composites rolled at initial temperature of 500 °C.

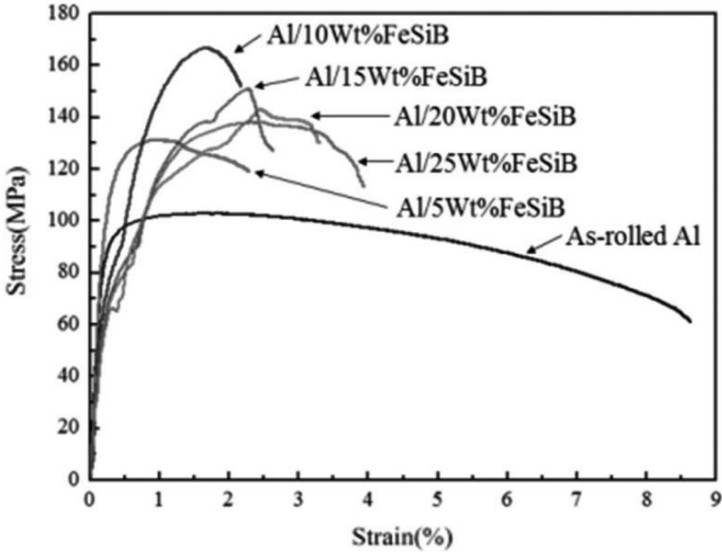

**Figure 2.** Tensile property curves of the roll bonding composite sheets with MG content of 5%, 10%, 15%, 20%, and 25% at initial temperature 450 °C and deformation 80%.

To visually observe the composite effect of the $Fe_{80}Si_9B_{11}$ MG strip-reinforced 1060 aluminum alloy–matrix composite, we used scanning electron microscopy (SEM) to observe the microdistribution of MG in the aluminum alloy matrix and microinterface bonding, as shown in Figure 3a, which shows an SEM image of the combination of the two microinterfaces at a content of the $Fe_{80}Si_9B_{11}$ MG of 5% in the matrix 1060 aluminum alloy. The strip is $Fe_{80}Si_9B_{11}$ MG, which is distributed in a flat shape, without large-area intermediate fracture, while the remaining part is 1060 aluminum alloy. Figure 3b shows the microdistribution of reinforcements in the matrix and a combination of the two microinterfaces in the composite material with an MG content of 10%. The MG strip and aluminum alloy matrix are closely combined, the lower interface boundary is clear, and there are no cracks and holes.

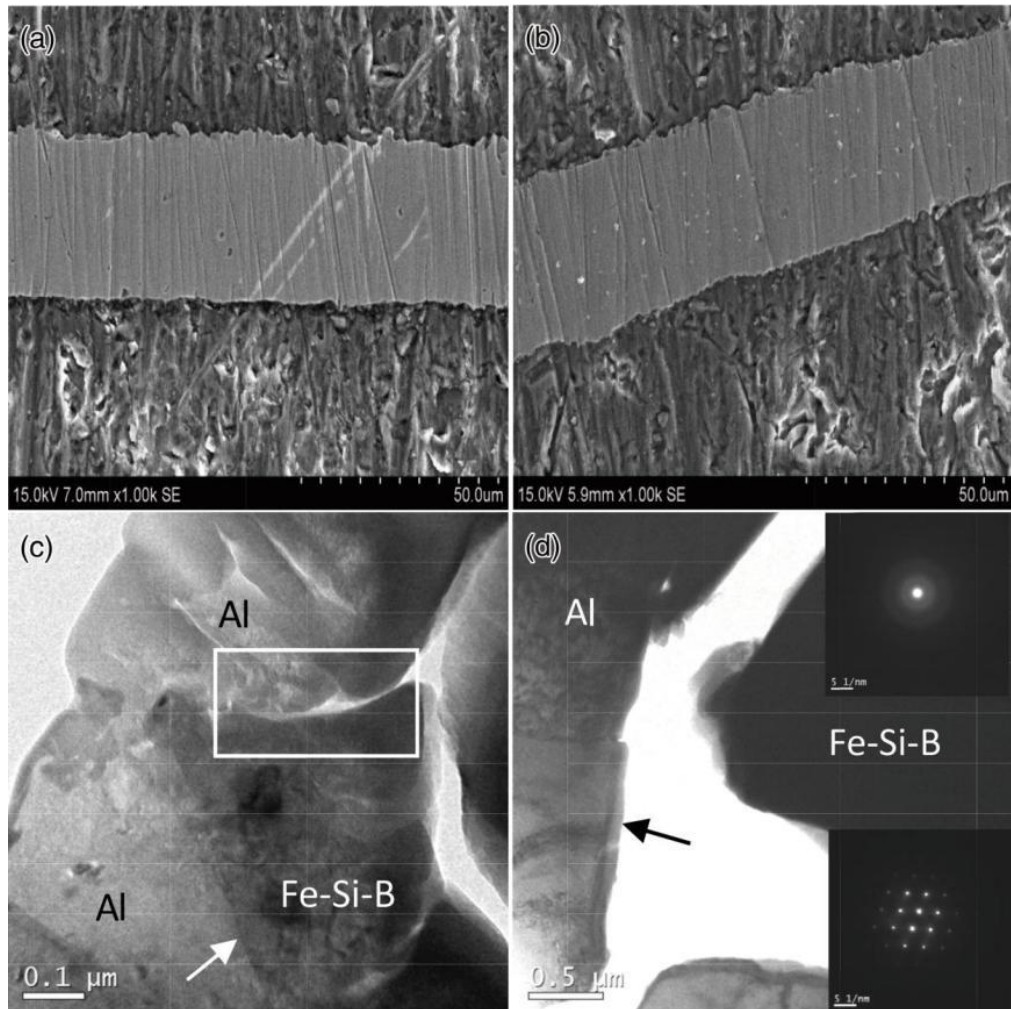

**Figure 3.** High- and low-magnification SEM photos and TEM bright field images of the Al/Fe$_{80}$Si$_9$B$_{11}$/Al composite plate after roll bonding at an initial temperature of 450 °C and deformation 80%. (**a**) Al/5% Fe$_{80}$Si$_9$B$_{11}$/Al; (**b**) Al/10% Fe$_{80}$Si$_9$B$_{11}$/Al; (**c,d**) Al/10%Fe$_{80}$Si$_9$B$_{11}$/Al.3.2. High-Pressure Torsion.

The presence or absence of metallurgical bonding was further evaluated by TEM, as shown in Figure 3c,d. The line of separation indicated by the white arrow in Figure 3c shows the bonding interface between the aluminum alloy and MG. The interface is tightly bonded without cracks and inclusions, which indicates that the two are well bonded in this area. However, it should not be ignored that, as shown in the white box in the figure, that there are obvious cracks between the aluminum alloy and MG. The cracks in the adjacent area even reach 0.05 μm. The interface in this area is not well bonded. Figure 3d shows a TEM image of another crack. The crack in this figure is larger than that in Figure 3c, and the bonding effect is worse. The line of separation indicated by the black arrow in Figure 3d shows the bonding interface between the aluminum alloy and MG. A small amount of MG adheres to the edge of one side of the aluminum alloy, perhaps because the rheological ability of the aluminum alloy and MG is inconsistent, resulting in the tearing of the aluminum alloy and MG during plastic deformation, leaving a small amount of MG on the aluminum alloy. This phenomenon indicates that there is a mechanical interface between the aluminum alloy and MG, but it does not form a real bond.

The microstructural observation of the Al/Fe$_{80}$Si$_9$B$_{11}$/Al composite synthesized by a single-pass rolling at 450 °C with a large reduction of 80% shows that the interface bonding of each group of samples is poor. Even if the interface is formed, it belongs to physical bite rather than metallurgical bonding, which is the main factor restricting the

mechanical properties of the composite. The composite material formed by rolling 10% MG and 1060 aluminum alloy is twisted one, two, three, four and five times under loading pressures of 0.55, 1.1, 1.65, and 2.2 GPa to fabricate composite discs. A nanoindentation test is carried out to determine the interface hardness of the sample. According to the equivalent strain formula, the accumulated strain at the near edge of the HPT specimen is largest, so the hardness is tested at the near edge. Each group of nanoindentation dot tests starts from the aluminum matrix, passes through the interface area, and ends at the MG. The distance between the test points and interval are same. Regarding the hardness distribution of nanoindentation under various loading pressures, the hardness of the 1060 aluminum alloy near the interface fluctuates around 800 MPa, while the hardness of the aluminum alloy increases slightly with the increase in the number of torsion turns, which is a result of work hardening. Under various loading pressures, when the number of torsion turns is small, the hardness values from the aluminum alloy to the MG are different, and there is no transition at the interface, which indicates that the bonding between the two regions is poor. Upon torsion at 0.55 GPa for six turns, torsion at 1.1 GPa for four turns, and torsion at 1.65 and 2.2 GPa for three turns, the change trend of the hardness near the interface has an "s" shape. The changes in hardness from the aluminum alloy with a lower hardness to the MG with a higher hardness are gradient-like, rather than cliff-like. There is also a transition zone where the hardness exceeds that of the MG. Figure 4 shows the hardness of the nanoindentation after four turns of torsion under a pressure of 1.1 GPa.

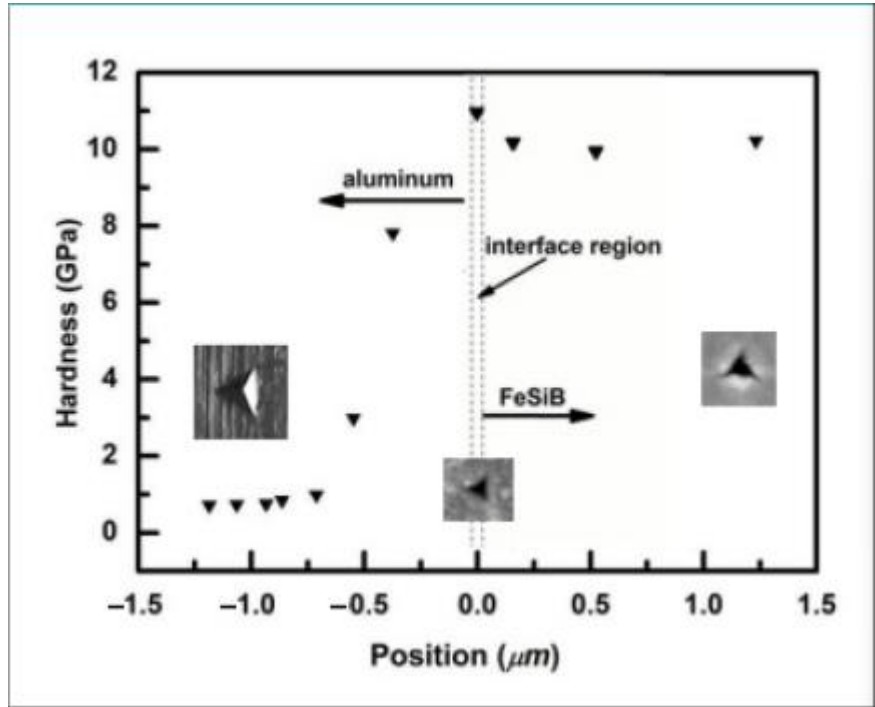

**Figure 4.** The hardness of the alloy around the bonding interface processed by HPT under 1.1 GPa at 4 torsion rounds.

The microstructure of the Al MG interface after HPT is observed by SEM and TEM on the samples twisted for one and four cycles under a pressure of 1.1 GPa. The results are shown in Figure 5a, which depicts a microscopy image of the interface of the sample twisted for one turn. The gray-white strip in the middle of the image is $Fe_{80}Si_9B_{11}$ MG, while the gray matrix on both sides is 1060 aluminum alloy. There are considerable gaps at the interface between the aluminum alloy and MG. In addition, many gray-white broken particles are mixed. Figure 5b shows a microscopy image of the specimen with four turns of torsion. There are no particle inclusions and cracks at the interface, and the boundary is clearly visible. Thus, a better combination of aluminum alloy and MG can be achieved

by four turns under an axial pressure of 1.10 GPa. A bright-field TEM image at the microinterface of the composite material is shown in Figure 5c. The dark part at the lower left side is $Fe_{80}Si_9B_{11}$ MG, while the bright part in the middle is the aluminum matrix. The interface between the two is very close, as indicated by the black arrow. Notably, no boundary seam can be observed; there is a certain transition zone at the interface. Figure 5d shows an HRTEM image near the interface, indicated by the black arrow in Figure 5c and the corresponding micro-area electron diffraction patterns of the different materials. Through the calibration of the diffraction patterns, it is determined that the $Fe_{80}Si_9B_{11}$ MG marked with halo ring diffraction patterns on the left side is in a glass state, while the 1060 aluminum alloy marked with dot diffraction patterns on the right side is in a crystalline state. The interface transition region between the two dotted lines shows a dot-like ring-shaped mixed diffraction pattern. The diffraction ring is wider than that of the iron-based amorphous structure, which indicates that the degree of ordering of amorphous atomic clusters is improved and that there is a tendency for crystallization.

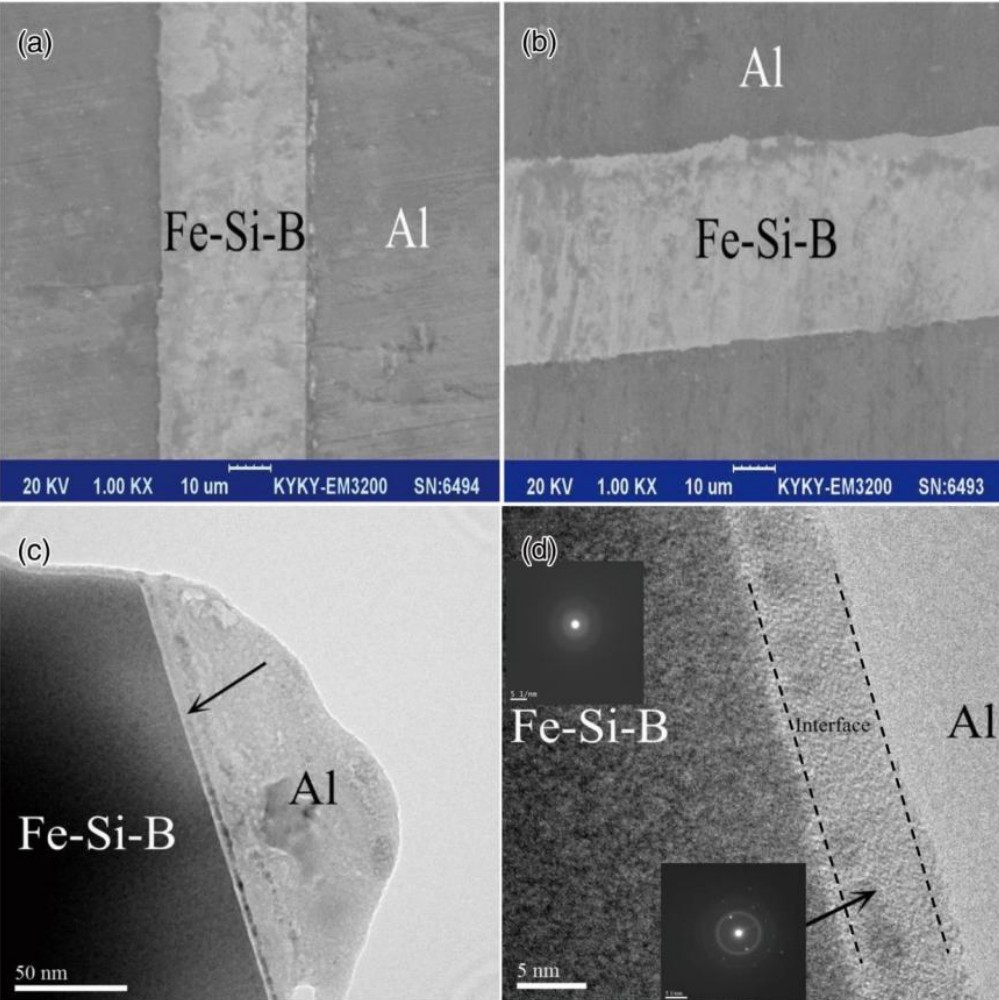

**Figure 5.** SEM photos of the interface of the composite wafer synthesized by high-pressure torsion at 10 GPa and TEM photo of the interface of the composite wafer synthesized by high-pressure torsion for 4 turns at 1.10 GPa. (**a**) Twist 1 turn; (**b**) twist 4 turns; (**c**) bright field image of the interface; (**d**) high-resolution image of the interface marking the selected area electron diffraction pattern (the crystalline characterization of Al can be seen in the below inset pattern of the selected area's electron diffraction, along with the amorphous halo in the same inset pattern).

## 4. Discussion

After the composite material with a MG content of 5% is rolled, a small amount of MG is distributed in the aluminum matrix in a flat shape. Although the large deformation of 80% can provide high rolling pressure, most of the MG acts on the aluminum alloy matrix to cause plastic deformation. The actual pressure of the MG reinforcement is considerably lower than the positive rolling pressure. According to the interface bonding theory proposed by N. Bay, because the synthetic pressure is too small, the coating on the surface of the reinforcement or substrate cannot be broken, the pure substrate cannot be exposed, and the phenomenon that the normal pressure will squeeze the pure substrate into the cracks of the coating on each other will not occur. The active surfaces of the two materials cannot meet, so that the true bonding cannot be formed. When the content of MG is increased to 10%, the positive pressure is increased, which provides a sufficient pressure for the bonding of the interface and ensures that the two form a solid bond, similar to the results of J. Ragani et al. [18], who combined an $Mg_{65}Cu_{25}Gd_{10}$ bulk amorphous structure with an aluminum alloy by the $C_O$ extrusion method, which is similar. However, the interface bonding is not ideal when the content of MG is high, likely as, under strong and large deformation, the brittle MG strip will not only contact and rub with the aluminum alloy matrix, but also produce collision, delivery, and other interactions between MGs. The aluminum alloy is a soft metal. Generally, when MG meets with it, the aluminum alloy can offset the excessive concentration of stress between the two through extension deformation, thus avoiding the large-scale occurrence of fracture to a certain extent. However, the MG is hard and brittle. When the two meet, under the rapid and high pressure application of the roller, the pressure cannot be released through an immediate deformation. The excessive stress concentration will inevitably lead to its breaking. The broken MG strip may change the original parallel arrangement with the flow of the aluminum alloy, and cross horizontally and vertically. If these fragments are sufficiently small, they may become a particle reinforcement phase of the aluminum alloy, which is conducive to improving the mechanical properties, including the high hardness of the matrix. However, most of the broken MG exists in the form of fragments. Therefore, when the content of MG is relatively small, the MG mainly interacts with the soft aluminum alloy matrix during the rolling. Due to the rolling force, the MG flows viscously with the plastic deformation of the aluminum alloy. It will not accumulate too- much stress, so that it will not cause too much brittle fracturing. Although the content is small in the aluminum matrix, it can still provide excellent characteristics, such as high strength and high hardness, which are conducive to improving the overall material strength. With the increase in the content of MG in the rolled composite plate, when the content reaches a certain level, the MG will not only be affected by the aluminum alloy matrix, but also may be affected by other MGs during the rolling. Under the high-speed extrusion shear deformation of the roller, the MG will flow with the aluminum alloy matrix and encounter the obstacles of other MG reinforcements. The two will converge, extrude, and stack together. The brittle MG is easily broken due to the rolling force. These fragments can easily become a defect source of mechanical failure, resulting in the reduction in its tensile strength. The TEM (Figure 3c,d) observation shows that there is no real physical metallurgical bond between the aluminum alloy and MG, which affects the mechanical properties of the composite.

According to Equation (1), the plastic deformation of the disc rim reaches the maximum upon HPT. Figure 4 shows the hardness of the alloy at the rim of the samples processed by HPT under 1.10 GPa in four torsion rounds. The hardness values of the region near the bonding interface are shown, with the distance value of the interface defined as zero. The hardness of the aluminum is shown to the left, and that of the MG strip is shown to the right. As can be seen, the hardness of aluminum remains at a relatively lower value of around 0.9 GPa, but presents an abrupt increase near the bonding interface. With the torsion rounds reaching four, the hardness of the interface region is 11.03 GPa, higher than that of the MG strip (10.10 GPa). Similar results in the hardness study were also obtained with various pressures and torsion strains in other HPT experiments. The unusual hardness

of the interface implies that a new phase different from normal crystalline aluminum and MG strip was formed by HPT under a certain condition.

To ascertain the structure of the bonding interface, microscopy studies were conducted on the HPT processed samples. Figure 5c presents the TEM bright field image of the alloys after the HPT process under 1.1 GPa and four rounds of torsion. From Figure 5c, a distinct and continuous bonding interface can be seen between the aluminum and MG strip, indicating that good bonding was achieved by this method. The electron diffraction pattern inset in Figure 5c reveals that the joint was formed mainly between the glassy FeSiB and crystalline aluminum. Judging from the difference between the diffraction halo of the MG and that of the interface region, the amorphous structure of the MG matrix near the interface region was partially ordered [19] to some extent. A further study of the interface affected by HRTEM is given in Figure 5d. There exist three different parts: the FeSiB MG region that presents disordered features, the interface region of around 8 nm wide (between the dash lines) and the crystalline aluminum region presenting an ordered structure. In the glassy FeSiB matrix of the interface region, the existence of aluminum presenting an amorphous structure can be seen, and a concentration gradient due to diffusion (near the dashed line) can be observed. Under the HPT process, the aluminum atoms diffuse into the amorphous FeSiB matrix and present an amorphous state. Outside of the FeSiB matrix, in the range of around 3 nm wide, the aluminum phase remains in an amorphous structure, which is probably formed due to the structural difference between the amorphous FeSiB matrix and the crystalline aluminum phases under severe deformation conditions [20]. The amorphous aluminum joint is formed by diffusion of the aluminum atoms into the amorphous FeSiB matrix upon HPT.

According to the TEM studies, the unusually high hardness of the interface region found by the nano-indenter can be understood. The amorphous aluminum layer formed by HPT bears higher hardness than glassy FeSiB does. On the other hand, the diffusion of aluminum atoms to the glassy FeSiB matrix will enhance the hardness of the alloy. From Figure 6, the precipitation of the bcc–Fe nanocrystals from the MG matrix can be seen (not limited to the circled part in Figure 6, showing partial ordering of the atoms), with grain sizes smaller than three nanometers, which to some extent cause the distortion of the amorphous matrix, and the very small grain sizes of the nanocrystals would block the movement of dislocations [21,22], which also increases the hardness of the amorphous FeSiB alloy. Moreover, Figure 6 shows that the nanocrystalline bcc–Fe were also deformed by HPT, and as seen in the circled area, the deformation of the nanocrystals would also contribute to the increase in the hardness of the interface region. The formation of the nanocrystals upon bonding also enhances the hardness of the interface region.

Macroscopic hardness tests reveal the bonding strength between the atoms, and could provide some structural features of a material. Diffusion or phase changes usually cause changes in the hardness of a certain phase, and vice versa; for a specific condition, hardness studies can provide some structural information. In this study, once the nano-hardness of the interface region of the deformed sample reach a certain value higher than that of FeSiB MG, atomic bonding between MG and crystalline aluminum can be considered to have been achieved.

By the bonding experiment of MG to aluminum under HPT and the following nano-hardness tests, the critical torsion strains needed to bond MG and aluminum under various pressures were determined, as shown in Figure 7. An exponential fit of the pressure and strain data is also given. The effect of pressure and strain upon HPT bonding is $p \geq 85.25\exp(-\varepsilon/9.5) + 0.25$. When the pressure and the strain value meet the above formula, atomic bonding between MG and aluminum can be achieved. For the pressure value of 0.55 GPa, with a strain up to 5.35, atomic bonding can be achieved. The result shows that with sufficient deformation strain, even at relatively lower pressures, MG can be bonded atomically to crystalline aluminum.

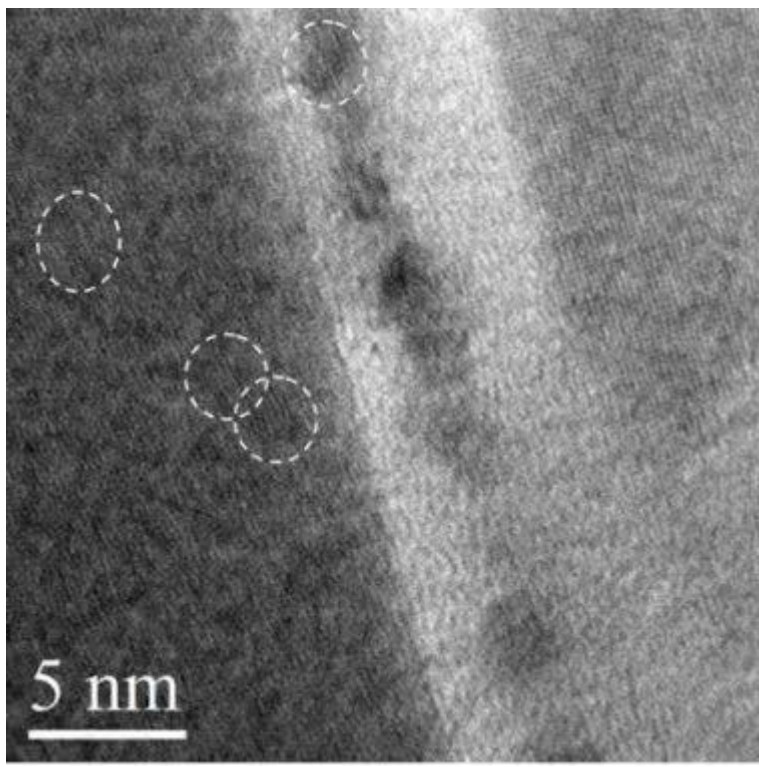

**Figure 6.** HRTEM image of the distortion of the NC Fe (partial ordering of the amorphous atoms are shown in the as circled part in Figure 6).

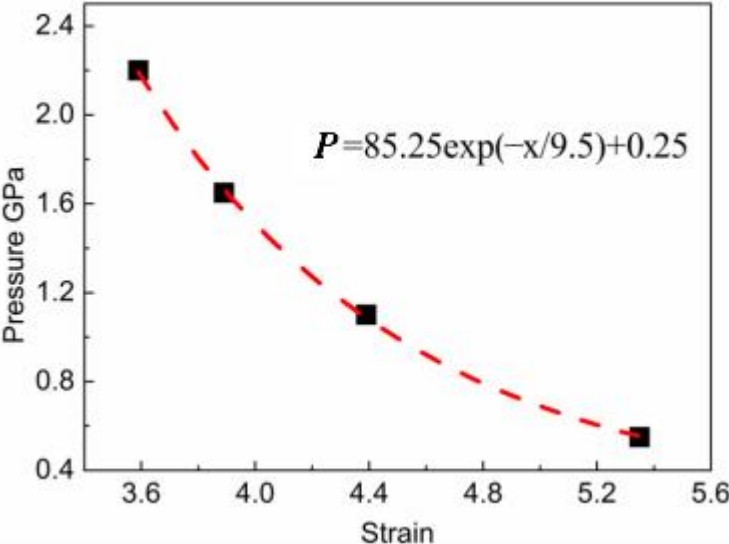

$$P=85.25\exp(-x/9.5)+0.25$$

**Figure 7.** The critical torsion strains under various pressures upon bonding and an exponential fit of the pressure and strain data.

In the TEM images shown in Figures 5 and 6, the microscopy observations show no shear bands or any clear evidence of severe deformation of the MG strip, which reveals that the MG strip and the ductile aluminum were not uniformly deformed. The bonding procedure of MG and aluminum upon HPT includes the following aspects: severe deformation of the aluminum, the relative movement between aluminum and MG strip at the interface, and the diminishing of the void between the joined couple. Following the diminishing of the voids, the surfaces of the joined couple achieved a greater contact area under high pressure, which favors the diffusion of the atoms.

The diffusions of atoms in MGes has been studied extensively; diffusion in MGes is a thermally activated [23], highly collective process [24,25]. In the present study, although processed at room temperature, the bonding interface achieved a temperature rise up to 120 K due to deformation under HPT [17]. On the other hand, an earlier study by Wang showed that under high pressures, the nano-crystallization temperature decreases with the increase in the loading pressure upon heating [26]. For the Zr-based MG, the nano-crystallization temperature decreases from 683 K at ambient pressure to 588 K at a pressure of 6 GPa, which also causes the decrease in the glass transition temperature under high pressure. Besides the high pressure and deformation effect, friction between MG and the aluminum upon HPT also causes a temperature rise and favors the diffusion of the aluminum atoms in the MG [27]. In this work, the formation of the nanocrystals in the FeSiB matrix (Figure 6) gives evidence that the alloy was heated to an elevated temperature to form nanocrystals upon the HPT process, which implies the collective diffusion of the aluminum atoms into the FeSiB alloy matrix. The collective diffusion layer up to 4 nm observed in the FeSiB matrix confirms the characteristics of the diffusion in MG. By explosive welding, Liu et al. [16] accomplished bonding between MG and crystalline Al in a very short time, and diffusion of the atoms was not observed, while the bonding couples kept their initial state. In this study, the bonding process was accomplished over a relatively longer time scale, and the thermally activated collective diffusion of aluminum into the MG matrix was promoted by the combined work of pressure and torsion deformation.

## 5. Conclusions

Aluminum matrix composites reinforced with $Fe_{80}Si_9B_{11}$ MG strips with different contents (5, 10, 15, 20 and 25%) were roll bonded at an initial temperature of 450 °C and the total deformation of 80%. X-ray detection analysis showed that the MG remained in an amorphous state and played the role of a reinforcement. The tensile strength of the composite with 10% MG reached 166 MPa, 62.7% higher than that of industrial pure aluminum, 102 MPa. TEM observations showed that the interface of the metal glass and aluminum remained in a state of mechanical bonding.

As to the HPT studies under different pressures, nano-indentation mechanical testing showed that the trend of the hardness near the interface presented an "S" shape, with a transition area. High-resolution transmission electron microscope observation showed that the width of the interface bonding layer between the MG and the aluminum alloy in the sample twisted four times under the pressure of 1.10 GPa was about 8 nm, which enabled the metallurgical bonding of the interface between MG and aluminum alloy.

Under different pressure and torsion conditions of 0.55 GPa, 1.10 GPa, 1.65 GPa, and 2.20 GPa, the cumulative equivalent strain variables required to achieve interface metallurgical bonding are 5.35, 4.39, 3.89, and 3.59, respectively. Therefore, the critical pressure and strain conditions required for the metallurgical bonding of industrial pure aluminum and Fe80Si9B11 MG interface are as follows:

$$P = 47.4\varepsilon^2 - 516.2\varepsilon + 1459.7 \tag{2}$$

**Author Contributions:** Conceptualization, S.S. and Y.J.; methodology, S.S.; software, S.S.; validation, X.Z. and H.G.; formal analysis, S.S.; investigation, S.S.; resources, Y.J.; data curation, S.S.; writing—original draft preparation, S.S.; writing—review and editing, S.S.; visualization, Y.J.; supervision, Y.J.; project administration, Y.J.; funding acquisition, Y.J. All authors have read and agreed to the published version of the manuscript.

**Funding:** This research was funded by NSFC (Grant No. 51471143).

**Institutional Review Board Statement:** Not applicable.

**Informed Consent Statement:** Not applicable.

**Data Availability Statement:** Not applicable.

**Acknowledgments:** The authors gratefully acknowledge J. W. Zhang for his insightful discussion of the TEM results.

**Conflicts of Interest:** The authors declare no conflict of interest.

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
