# Peer review of "Physical Metallurgical Bonding Layer Formed between Fe80Si9B11 Metallic Glass and Crystalline Aluminum in Rolled Composite Plate by High-Pressure Torsion at Room Temperature"

_metals, doi:10.3390/met12111929_

Round 1

Reviewer 1 Report

Moderate English changes are required. Please, check carefully.

Author Response

Thank the reviewers for their valuable suggestions on my article. We have already made the following corrections.

The authors’ responses to reviewer’s comments

Reviewer 1

Moderate English changes are required. Please, check carefully.

Response:

The manuscript is checked, following the reviews suggestion.

Reviewer 2 Report

The authors study aluminum matrix composites reinforced with Fe80Si9B11 metallic glass strips with different weight contents (5, 10, 15, 12 20 and 25%). After further high-pressure torsion under different pressures, nano-indentation mechanical test shows that the change trend of the hardness value near the interface is an ‘S’ shape with a transition area. High-resolution transmission electron microscope observation showed that the width of the interface bonding layer between metallic glass and aluminum alloy in the sample twisted for 4 turns under the pressure of 1.10GPa was about 8 nm, which realized the physical metallurgical bonding of the interface between metallic glass and aluminum alloy, rather than mechanical bonding by roll-bonding. Combined with other conditions, the cumulative equivalent strain variables required to realize the physical metallurgical bonding of the interface are calculated, and the critical stress-strain conditions required for the real bonding of the interface are defined. 

This is an interesting manuscript on the very relevant topic of metal-metallic glass (M-MG) interfaces, and their effects on mechanical properties. The authors shall address the following issues with the manuscript, before publication:

1. Since HPT was applied after the manufacturing of the interfaces, isn't it expected that the M-MG interface is not straight most of the time in the samples? The interface in Fig.3 seems remarkably straight, how is this possible? Can the authors show the interface structures (multiple interfaces in a single image, larger scale) at different levels of HPT? How the pattern changes with HPT?

2. Are the indents of Fig.4 done in a straight line? Or did they perform an array of indents at some spacing, and then selected particular points? Can the authors show the image of indents, and possibly which ones were selected for this particular plot (Fig.4)? The authors can show the indents' image next to Fig.4.

3. Can the authors add some comments in the caption of Fig.6, regarding the circling of precipitate grains, it will be beneficial to readers.

4. Do the authors have any understanding of the relationship shown in the Conclusions, with respect to dislocation theory and phenomenology? 

Author Response

Thank the reviewers for their valuable suggestions on my article. We have already made the following corrections.

The authors’ responses to reviewer’s comments

Reviewer 2

  1. Since HPT was applied after the manufacturing of the interfaces, isn't it expected that the M-MG interface is not straight most of the time in the samples? The interface in Fig.3 seems remarkably straight, how is this possible? Can the authors show the interface structures (multiple interfaces in a single image, larger scale) at different levels of HPT? How the pattern changes with HPT?

Response:

The M-MG interface is mainly determined by the shape of the metallic glass ribbon, because the glassy alloys bear higher strength than Al. under HPT processing, Al is easily deformed, the glassy ribbon could be broken into smaller pieces, but won’t be deformed, thus showed the straight interface. In this study, the HPT samples were simple sandwich structured with a single glassy ribbon for studying of the bonding, the case of multiple interface was not involved in HPT study.

  1. Are the indents of Fig.4 done in a straight line? Or did they perform an array of indents at some spacing, and then selected particular points? Can the authors show the image of indents, and possibly which ones were selected for this particular plot (Fig.4)? The authors can show the indents' image next to Fig.4.

Response:

The indent tests were made from the Al to the interface, but not exactly straightly or evenly point to point in the very limited space near the interface. The indents images are added in Fig.4.

  1. Can the authors add some comments in the caption of Fig.6, regarding the circling of precipitate grains, it will be beneficial to readers.

Response:

Comments are added in the caption of Fig.6

  1. Do the authors have any understanding of the relationship shown in the Conclusions, with respect to dislocation theory and phenomenology?

Response:

Dislocation related facts were rarely observed in this case, thus dislocation theory was not mentioned in this case. Understandings of the relationship between the bonding of M-MG were mainly discussed with respect to phenomenology between the HPT parameters and micro-hardness near the bonding interface.

Reviewer 3 Report

Dear S.Shan et al,

This paper investigates tensile properties and microstructures of hot-rolled aluminum composites with 5-25wt% FeSiB metallic glass, and hardness change and microstructures around bonding interface of the further high-pressure tensioned ones. The obtained results seem to be probable, but the following points should be amended for further improvement of this paper.

1.     The authors are questioning the impact of the hard metal layer on the mechanical properties of the composite materials in 1. Introduction (P2L55-59), but no answer to this question is given in this paper based on their experimental results. Does the newly formed bonding interface with a hardness of 11.03GPa increase the mechanical properties of the HPT processed discs? The reviewer wonders if hardness change around the bonding interface (Fig.4) does not necessarily ensure the positive impact, as the authors expect.

2.     The XRD curve of the original Fe90Si9B11 foil strip should be superimposed in Fig.1 to ensure “ The Fe90Si9B11 metallic glass remained in the amorphous state (P3L96).

3.     No Table 1 is given in this paper although the authors describe “The tensile strength data are shown in Table 1 (P3L110-111).

4.     No diffraction pattern is given in Fig.5(d) for ensuring “the 1060 aluminum alloy marked with dot diffraction patterns on the right side is in a crystalline state (P6L190-192)”.

Author Response

Thank the reviewers for their valuable suggestions on my article. We have already made the following corrections.

The authors’ responses to reviewer’s comments

Reviewer 3

1、The authors are questioning the impact of the hard metal layer on the mechanical properties of the composite materials in 1. Introduction (P2L55-59), but no answer to this question is given in this paper based on their experimental results. Does the newly formed bonding interface with a hardness of 11.03GPa increase the mechanical properties of the HPT processed discs? The reviewer wonders if hardness change around the bonding interface (Fig.4) does not necessarily ensure the positive impact, as the authors expect.

Response:

The authors did not mean to question“the impact of the hard metal layer on the mechanical properties of the composite materials” . the manuscript is focused on the roll bonding of Al and metallic glass (MG) strips and the Metallurgical bonding of MG to Al. in order to avoid misunderstanding, the above mentioned content is deleted. The newly formed interface just caused the increase of the hardness near the interface.

  1. The XRD curve of the original Fe90Si9B11 foil strip should be superimposed in Fig.1 to ensure “ The Fe90Si9B11 metallic glass remained in the amorphous state (P3L96).

The XRD curve of the original strip were the same as that of the strip after roll bonding, no difference can be examined by XRD, the description is given in the manuscript.

  1. No Table 1 is given in this paper although the authors describe “The tensile strength data are shown in Table 1 (P3L110-111).

Response

“The tensile strength data are shown in Table 1” is deleted

4、No diffraction pattern is given in Fig.5(d) for ensuring “the 1060 aluminum alloy marked with dot diffraction patterns on the right side is in a crystalline state (P6L190-192)”.

Response:

The crystalline characterization of Al can be seen in the below inset pattern of the selected area electron diffraction, along with the amorphous halo in the same inset pattern. This part of description is added in the figure caption.

Round 2

Reviewer 3 Report

None